# THE NORMALIZED FLOAT TRICK.
# NUMERICAL STABILITY FOR PROBABILISTIC
# CIRCUITS WITHOUT THE LOGSUMEXP TRICK

## ABSTRACT

Probabilistic circuits (PCs) are a class of tractable deep probabilistic models that compute event probabilities by recursively nesting sum and product computations. Unfortunately, this is numerically unstable. To mitigate this numerical stability issues, PCs are usually evaluated in log-space via the LogSumExp trick. In this paper we present an alternative to the ubiquitous LogSumExp trick, which we dub *normalized float trick*. Experimentally, we show that by simply changing the scheme guaranteeing numerical stability (from the LogSumExp to the normalized float trick) we can consistently and considerably boost the performance of PCs on common density estimation benchmarks,

## 1  INTRODUCTION

Probabilistic circuits (PCs) (Darwiche, 2003; Poon & Domingos, 2011) are a class of machine learning models that perform density estimation by constructing deeply nested mixture models. That is, they recursively nest mixture models in order to construct joint probability distributions. This has two advantages. Firstly, the so-obtained model is more expressive than usual shallow mixture models. Secondly, PCs retain interesting properties with regard to tractability, such as any-order marginalization. Concretely, consider the probability distribution $p(x_1, \ldots, x_N)$ expressed as a circuit. The (hierarchical) mixture model nature then allows us to marginalize out in poly-time (in size of the circuit) any variable $x_i$, i.e. to compute in poly-time the following marginal $p(x_1, \ldots, x_{i-i}, x_{i+i}, \ldots x_N)$. We give a graphical representation of a probabilistic circuit in Figure 1 (left).

From a numerical stability perspective, deeply nesting mixture models pose a problem, however. When evaluation a circuit we end up repeatedly multiplying floating point numbers with each other that lie in the unit interval $[0, 1]$. The problem is then that this quickly results into numerical underflow. For this reason virtually all implementations of probabilistic circuits use the log-domain to perform probabilistic inference (Peharz et al., 2020; Liu et al., 2024). However, naively performing computations in the log-domain does not entirely resolve numerical stability issues. A common technique to resolve this is the LogSumExp trick.[1]

In this paper we introduce an alternative to the LogSumExp trick that we dub the *normalized float* trick or NoFlo trick and use it to evaluate probabilistic circuits. A curious experimental finding is that using our new NoFlo trick instead of the usual LogSumExp trick leads to improved density estimation capabilities of probabilistic circuits – simply by changing the numerically stable computation scheme.

The remainder of the paper is organized as follows. We first give the necessary preliminaries on probabilistic circuits (Section 2). We then introduce the NoFlo trick and contrast it to the LogSumExp trick in the context of PCs (Section 3). In Section 4 we give experimental evidence that the NoFlo trick achieves consistently better density estimation than the LogSumExp trick on a suite of density estimation benchmarks. We end the paper with concluding remarks in Section 5.

---

[1] As the LogSumExp trick is a widely known method for performing computationally stable probabilistic inference we only discuss it in more detail in Appendix B.

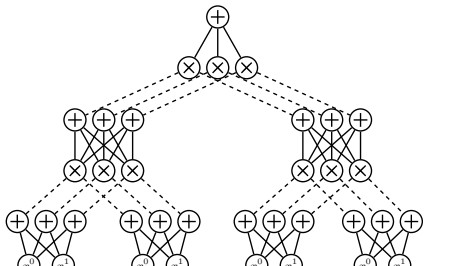 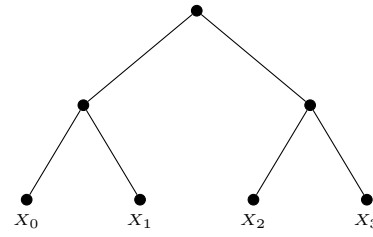

Figure 1: Left: Layered PC over four binary variables $X_i$ with $i \in \{0, 1, 2, 3\}$ taking values $x_i^0$ or $x_i^1$. Within each partition in the bottom layer three mixtures for each of the four $X_i$'s are constructed using weighted sums (weights are not shown in the graphical representation but are present on the edges feeding into sum units). The sum and product units in the circuit are the elemental computing units and make constitute the partitions. The circuit in Figure 1 has, except at the very top and bottom, three components in each partition. As increasing the number of components per partition increases the number of parameters (weights on edges feeding into sum units), the number of components per partition controls the capacity of a circuit. Right: partition tree abstracting the layered PC on the left.

## 2 PROBABILISTIC CIRCUITS AS PARTITION CIRCUITS

Inspired by simple feed-forward neural networks, Peharz et al. (2019; 2020) introduced the concept of layered probabilistic circuits These layered circuits are amenable to trivial parallelization and can be run on modern discrete GPUs and constitute the de facto standard approach towards constructing probabilistic circuits. In Figure 1 we give a graphical representation of such a layered circuit where we follow the construction introduced by Shih et al. (2021). For the sake of conciseness we describe here only so-called *structured decomposable and smooth* PCs. For a broader overview we refer the reader to the excellent work of Vergari et al. (2021).

Layers within a layered PC constitute blocks of computational units that are processed sequentially in a bottom-up fashion. Layers consist themselves of so-called partitions. The circuit in Figure 1 has four partitions in the leaf layer (very bottom), two in the subsequent sum and product layers, and a single partition in the last product layer and in the final root node (top most sum). By construction, partitions in the same layer have disjoint scopes. That is, they are functions over disjoint sets of variables. This property is called structured decomposability in the PC literature (Darwiche, 2011). Note also that layered PCs are by construction smooth (Darwiche, 2001): the union of the scopes of the partitions within a layer is exhaustive. That is, the union of the scopes equals the set of variables given as input to the circuit.

A (layered) PC is then parametrized by weighing the inputs to the sum units with positive real numbers – giving rise to an unnormalized probability distribution over the input variables. Due to the properties of (structured) decomposability and smoothness the distribution can be normalized in time polynomial in the size of the circuit (Peharz et al., 2015). Thanks to the properties of smoothness and (structured) decomposability we can also marginalize out single variables from a PC. Again in time polynomial in the size of the circuit.

Recently, Zuidberg Dos Martires (2024) introduced the concept of *partition trees* to abstract away certain aspects of smooth and structured decomposable probabilistic circuits. We give such a partition tree on the right side of Figure 1. We now take this abstraction a step further and define the computations performed by the probabilistic circuit not on the atomic computation units of the circuit itself (i.e. sum and product units) but use the nodes in the partition tree as the elemental computation units. In other words, we regard the partition tree as a computation graph. To make this distinction explicit we dub these computation graphs *partition circuits*.

**Definition 2.1** (Partition Circuit). *A partition circuit over a set of variables is a parametrized computation graph taking the form of a binary tree. The partition circuit consists of three kinds of computation units: leaf and internal units, as well as a single root. Units at the same distance from the root form a layer. Furthermore, let $\pi_k$ denote the root unit or an internal unit. The unit $\pi_k$ then receives its inputs from two units in the previous layer, which we denote by $\pi_{k_l}$ and $\pi_{k_r}$. Each*

*computation unit is input to exactly one other unit, except the root unit, which is the input to no other unit.*

**Definition 2.2** (Layered Probabilistic Circuit). *Let* $\mathbf{X} = \{X_0, \ldots, X_{M-1}\}$ *(taking values* $\mathbf{x} = \{x_0, \ldots, x_{M-1}\}$*) be a set of categorical random variables. Furthermore, let C be a positive integer denoting the number of components. We define a layered probabilistic circuit as a partition circuit whose computation units take the following functional form:*

$$\pi_k(\mathbf{x}_k) = \begin{cases} W_k \times one\_hot(x_k), & W_k \in \mathbb{R}_{\geq 0}^{C \times M} & \text{if } k \text{ leaf unit, i.e. } \mathbf{x}_k = \{x_k\} \\ W_k \times \left(\pi_{k_l}(\mathbf{x}_{k_l}) \odot \pi_{k_r}(\mathbf{x}_{k_r})\right), & W_k \in \mathbb{R}_{\geq 0}^{C \times C} & \text{if } k \text{ internal unit} \\ W_k \times \left(\pi_{k_l}(\mathbf{x}_{k_l}) \odot \pi_{k_r}(\mathbf{x}_{k_r})\right), & W_k \in \mathbb{R}_{\geq 0}^{1 \times C} & \text{if } k \text{ root unit.} \end{cases} \quad (1)$$

*Here we use the symbols* $\times$ *to denote the matrix product and Hadamard product, respectively. Additionally, we necessitate the matrices* $W_k$ *to be row-normalized. That is,* $\forall k, i : \sum_j W_{kij} = 1$*, where the i and j indices index the matrix.*

Note that in the definition above we used Hadamard products to fold computations from the previous layer. This is also called canonical polyadic layer (Carroll & Chang, 1970) and the de facto standard in the PC literature (Shih et al., 2021; Liu & Van den Broeck, 2021). For the sake of exposition we will limit ourselves to canonical polyadic layers and refer the reader to (Mari et al., 2023) for alternatives.

**Proposition 2.3.** *Layered PCs are valid probability distributions.*

*Proof.* The proof follows a similar structure to proving that probabilistic circuits with explicit sum and product nodes are valid probability distributions. For the sake of completeness we provide this proof for PCs as partition circuits in Appendix A $\qquad\square$

## 3 THE NORMALIZED FLOAT TRICK FOR PROBABILISTIC CIRCUITS

### 3.1 THE LOGSUMEXP TRICK FOR PROBABILISTIC CIRCUITS

In order to guarantee numerical stability probabilistic circuits are usually evaluated in log-space. This means that we evaluate layers of the circuits by computing log-probabilistic instead of (linear) probabilities. We briefly re-derive the computations performed in the internal units such that they output indeed log-probabilities.

$$\pi_k = W_k \times \left(\pi_{k_l} \odot \pi_{k_r}\right) \Leftrightarrow \log \pi_k = \log\left(W_k \times \left(\pi_{k_l} \odot \pi_{k_r}\right)\right) \quad (2)$$

$$\Leftrightarrow \lambda_k = \log\left(W_k \times \left(\exp \lambda_{k_l} \odot \exp \lambda_{k_r}\right)\right) \quad (3)$$

$$\Leftrightarrow \lambda_k = \log\left(W_k \times \exp\left(\lambda_{k_l} + \lambda_{k_r}\right)\right), \quad (4)$$

where we omitted the dependency of $\pi_k$ on $\mathbf{x}_k$. Note also that $\pi_k \in \mathbb{R}_{\geq 0}^C$ and that the logarithm and exponential are applied element-wise on the vectors. Unfortunately, the right-hand side of Equation 4 is numerically unstable, and we need to adopt the LogSumExp trick:

$$\lambda_k = \log\left(W_k \times \exp\left(\lambda_{k_l} + \lambda_{k_r} - \alpha_k + \alpha_k\right)\right) \quad (5)$$

$$\Leftrightarrow \lambda_k = \log\left(W_k \times \exp\left(\lambda_{k_l} + \lambda_{k_r} - \alpha_k\right)\right) + \alpha_k, \quad (6)$$

where $\alpha_k \in \mathbb{R}$, with $\alpha_k = \max_i p_{ki}$ being the standard choice. Here, $i$ indexes the vector elements. Equation 6 now tells us how to compute the log-probability ($\lambda_k$) in the current layer $k$ given the incoming log-probabilities ($\lambda_{k_l}$ and $\lambda_{k_r}$) in a numerically stable fashion.

### 3.2 INTRODUCING THE NOFLO TRICK

We now propose an alternative to the LogSumExp trick: the NoFlo trick. The basic idea is to represent a probability vector, i.e. a vector whose entries fall into the $[0, 1]$ interval using a normalized

probability vector and an extra scalar:

$$\pi_k = \underbrace{\frac{\pi_k}{\max_i \pi_{ki}}}_{=:\hat{\pi}_k} \exp\underbrace{\left(\log\max_i \pi_{ki}\right)}_{=:\beta_k} \tag{7}$$

$$= \hat{\pi}_k e^{\beta_k} \tag{8}$$

This means we identify each $\pi_k$ via the vector $\hat{\pi}_k$ and the scaling factor $\exp\beta_k$. We now apply this decomposition to the expression for computing a layer in a probabilistic circuit:

$$\pi_k = W_k \times \left(\pi_{k_l} \odot \pi_{k_r}\right) \Leftrightarrow \hat{\pi}_k e^{\beta_k} = W_k \times \left(\hat{\pi}_{k_l} e^{\beta_{k_l}} \odot \hat{\pi}_{k_r} e^{\beta_{k_r}}\right) \tag{9}$$

$$= \underbrace{W_k \times \left(\hat{\pi}_{k_l} \odot \hat{\pi}_{k_r}\right)}_{=:\tilde{\pi}_k} e^{\beta_{k_r} + \beta_{k_r}} \tag{10}$$

$$= \tilde{\pi}_k e^{\beta_{k_r} + \beta_{k_r}} \tag{11}$$

Here we have to be careful now as computing $\tilde{\pi}_k$ consists of multiplying (and adding) together numbers in the $[0, 1]$ interval. If this is done repeatedly, for instance during the evaluation of the various layers in a probabilistic circuit, we risk again to run into underflow issues. For this reason we normalize $\tilde{\pi}_k$ again:

$$\pi_k = \frac{\tilde{\pi}_k}{\gamma_k} e^{\beta_{k_r} + \beta_{k_r} + \log\gamma_k}, \tag{12}$$

with $\gamma_k := \max_i \tilde{\pi}_{ki}$.

We can now compute at each layer a representation for $\pi_k$ by decomposing it into the product $\hat{\pi}_k e^{\beta_k}$ and identifying $\hat{\pi}_k$ with $\tilde{\pi}_k / \gamma_k$ and $\beta_k$ with $\beta_{k_r} + \beta_{k_r} + \log\gamma_k$. This then simply means we need to compute at each unit in the PC the following two equations:

$$\hat{\pi}_k = \frac{\tilde{\pi}_k}{\gamma_k} \tag{13}$$

$$\beta_k = \beta_{k_r} + \beta_{k_r} + \log\gamma_k \tag{14}$$

Finally, at the root node we are then usually interested in the log-probability $\lambda_{root} = \log\pi_{root}$ instead of the two quantities $\hat{\pi}_{root}$ and $\beta_{root}$. Fortunately, we can readily resolve this issue:

$$\lambda_{root} = \log\pi_{root} = \log\left(\hat{\pi}_{root} e^{\beta_{root}}\right) \tag{15}$$

$$= \log\hat{\pi}_{root} + \beta_{root}, \tag{16}$$

where $\hat{\pi}_{root}$ and $\beta_{root}$ are computed using Equations 13 and 14. We give the pseudocode to compute $\lambda_{root}$ using the NoFlo trick in Algorithm 2 and the for the evaluation with the LogSumExp trick in Algorithm 1.

### 3.3 Estimating the Relative Computational Costs

Using Algorithm 1 and Algorithm 2 we can also estimate the computational cost of both approaches towards numerical stability. To this end we estimate the cost needed to perform the computations present in an internal unit. We denote these costs by $\kappa_{\text{LogSumExp}}$ and $\kappa_{\text{NoFlo}}$, respectively.

When using the LogSumExp trick, we first compute the element-wise addition of two vectors of length $C$, which gives us a cost of $C \cdot \kappa_{add}$, cf. Line 5, Algorithm 1. We then obtain the maximum value of the resulting vector by performing $C$ comparisons, cf. Line 5, Algorithm 1. Subsequently, the maximum value in subtracted from each individual element, allowing us to take the element-wise exponential before performing a matrix-vector multiplication. Finally, we take the element-wise logarithm and add the maximum value back to each element of the vector. We can hence estimate the cost of performing the computations in an internal unit of a PC using the LogSumExp trick as follows:

$$\kappa_{\text{LogSumExp}} = C \cdot \kappa_{add} + C \cdot \kappa_{comp} + C \cdot \kappa_{sub} + C \cdot \kappa_{exp} + \tag{17}$$

$$C \cdot (C \cdot \kappa_{mul} + C \cdot \kappa_{add}) + \tag{18}$$

$$C \cdot \kappa_{log} + C \cdot \kappa_{add}, \tag{19}$$

**Algorithm 1** Circuit Evaluation with LogSum-Exp

**Input:** set of computation units $\mathbf{K}$, random variable values $\mathbf{x}$

**Output:** $\lambda_{root}(\mathbf{x})$
1: **for** $k$ **in** topological_sort($\mathbf{K}$) **do**
2:     **if** $k$ is leaf unit **then**
3:         $\lambda_k \leftarrow \log \left[ W_k \times \text{one\_hot}(x_k) \right]$
4:     **else**   ▷ $k$ is internal unit or root unit
5:         $\lambda_k \leftarrow \lambda_{k_l} + \lambda_{k_r}$
6:         $\alpha \leftarrow \max_i \lambda_{ki}$
7:         $\lambda_k \leftarrow \log \left[ W_k \times \exp(\lambda_k - \alpha) \right] + \alpha$
8:     **end if**
9: **end for**
10: **return** $\lambda_{root}$

**Algorithm 2** Circuit Evaluation with NoFlo

**Input:** set of computation units $\mathbf{K}$, random variable values $\mathbf{x}$

**Output:** $\lambda_{root}(\mathbf{x})$
1: **for** $k$ **in** topological_sort($\mathbf{K}$) **do**
2:     **if** $k$ is leaf unit **then**
3:         $\hat{\pi}_k \leftarrow \log \left[ W_k \times \text{one\_hot}(x_k) \right]$
4:         $\gamma \leftarrow \max_i \hat{\pi}_{ki}$
5:         $\hat{\pi}_k \leftarrow \hat{\pi}_k \cdot {}^1\!/\gamma$
6:         $\beta_k \leftarrow \log \gamma$
7:     **else**   ▷ $k$ is internal unit or root unit
8:         $\hat{\pi}_k \leftarrow W_k \times \left( \hat{\pi}_{k_l} \odot \hat{\pi}_{k_r} \right)$
9:         $\gamma \leftarrow \max_i \hat{\pi}_{ki}$
10:       $\hat{\pi}_k \leftarrow \hat{\pi}_k \cdot {}^1\!/\gamma$
11:       $\beta_k \leftarrow \beta_{k_l} + \beta_{k_l} + \log \gamma$
12:     **end if**
13: **end for**
14: **return** $\log \hat{\pi}_{root} + \beta_{root}$

Table 1: Clock cycles needed for different floating point operations using the x86 instruction set. Numbers retrieved from (Agner, 2022).

| FADD addition | FSUB subtraction | FMUL multiplication | FDIV division | FYL2X $y \log_2 x$ | F2XM1 $2^x - 1$ | FUCOMI comparison |
|---|---|---|---|---|---|---|
| $\kappa_{add}$ | $\kappa_{sub}$ | $\kappa_{mul}$ | $\kappa_{div}$ | $\kappa_{log}$ | $\kappa_{exp}$ | $\kappa_{comp}$ |
| 3 | 3 | 5 | 19 | 29 | $\approx 68$ | 2 |

where the expression in Equation 18 describes the cost of performing the matrix-vector product. Note that as $C$ grows the cost will be dominated by the matrix-vector product, due to its quadratic dependency on $C$.

An analogous analysis of Algorithm 2 gives us the following expression for the computational cost of evaluating an internal circuit unit using the NoFlo trick:

$$\kappa_{\text{NoFlo}} = C \cdot \kappa_{mul} + \tag{20}$$
$$C \cdot (C \cdot \kappa_{mul} + C \cdot \kappa_{add}) + \tag{21}$$
$$C \cdot \kappa_{comp} + \cdot \kappa_{div} + C \cdot \kappa_{mul} + \tag{22}$$
$$\kappa_{log} + 2C \cdot \kappa_{add}, \tag{23}$$

with the expression in Equation 21 describing again the cost for the matrix-vector multiplication.

In order to render $\kappa_{\text{LogSumExp}}$ and $\kappa_{\text{NoFlo}}$ commensurable we estimate the cost of the individual operations, e.g. $\kappa_{add}$ via the clock cycles needed to compute them. We summarize the clock cycles for the different operations in Table 1. This gives us the following cost estimates:

$$\kappa_{\text{LogSumExp}} \approx \underbrace{8C^2}_{matrix-vector} + 108C \qquad \kappa_{\text{NoFlo}} \approx \underbrace{8C^2}_{matrix-vector} + 18C + 48. \tag{24}$$

We see that in terms of clock cycles the NoFlo trick is slightly cheaper but also that the cost for both is (unsurprisingly) dominated by the matrix-vector product for larger $C$. Both approaches do also have a similar memory footprint with the NoFlo being slightly more expensive in this regard as do not only need to store a vector of size $C$ but also an extra float $\beta_k$.

Note that these are extremely rough estimates and should only be used to estimate the order of magnitude of the cost. Note also, that modern hardware usually has also access SIMD instructions (on CPU) and parallelizable kernels (on GPU), allowing for performing matrix-vector multiplications in parallel. Furthermore, we did not take into account any cost associated with reading from and writing to memory.

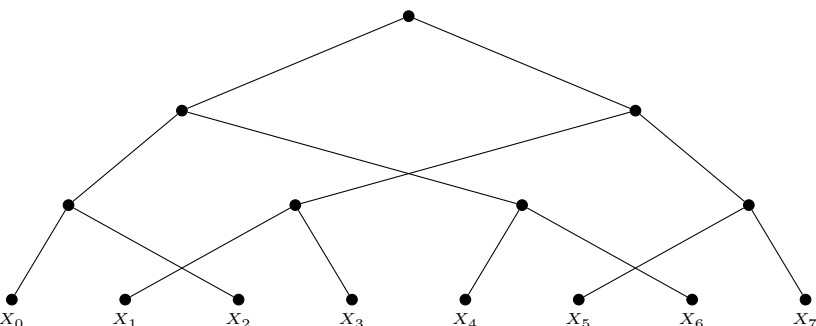

Figure 2: Graphical representation of the partition tree for eight variables using the `cross` structure.

## 4  EXPERIMENTAL EVALUATION

For the experimental evaluation we implemented probabilistic circuits using the NoFlo trick as well as the LogSumExptrick in PyTorch. We set up our experiments in Lightning[2] and ran them on a DGX-2 machine with V100 Nvidia cards. Our experimental evaluation is performed using the MNIST family of datasets. That is, the original MNIST (Deng, 2012), FashionMNIST (Xiao et al., 2017), and also EMNIST (Cohen et al., 2017).

### 4.1  DENSITY ESTIMATION WITH THE NOFLO AND LOGSUMEXP TRICKS

We compare the density estimation capabilities of PCs using the NoFlo trick to the ones using the traditional LogSumExp trick. To this end we encode gray-scale pixels by associating each of the $256$ possible pixel values to the outcome of a categorical random variable. For MNIST this means that we have $28 \times 28$ categorical random variables. These form the leaves of the probabilistic circuits.

We then construct two circuits with two distinct structures. The first structure, which we dub `neighbor` merges in an alternating fashion neighboring rows and columns of the 2D image. Specifically, in the first layer rows 0 and 1 get merge and rows 2 and 3 and so on. In the next layer the columns get merged. This pattern repeats until only a single partition is left, and we reach the root. In the case of layers with an uneven number of rows/columns we also allow for merging three neighboring rows/columns

For the second circuit structure, to which we will refer as `cross`, we merge rows/columns by merging consecutive evenly and unevenly numbered rows/columns. This means, for instance, that in the first layer rows 0 and 2 are merged, rows 1 and 3, rows 4 and 6, and so on. In alternating layers we then alternate between merging rows and columns. We exemplify the `cross` structure in Figure 2 for the 1D case.[3]

We then train both circuit structures by maximizing the log-likelihood using, on the one hand the NoFlo trick ,and on the other hand the LogSumExp trick. We compare the two schemes for numerical stability using the *bits per dimension* metric, which is calculated from the average negative log-likelihood ($\overline{NLL}$) as follows: $bpd = \overline{NLL}/(\log 2 \times D)$ ($D = 28^2$ for MNIST datasets).

Furthermore, we used the Adam optimizer (Kingma & Ba, 2014) with a learning rate of $0.05$ and otherwise default hyperparameters. The batch size was set to $50$ and the maximum number of epochs was set to $50$ as well. The best model was selected using a $95 - 5$ train-validation data split where we monitored the negative log-likelihood on the validation set. The validation split was also used for early stopping: training stopped preemptively when there has not been any improvement on the validation set for five epochs.

We report our results in Tables 2 (`neighbor`) and Table 3 (`cross`), where we can see that independent of the structure used (Table 2 vs. Table 3), independent of any of the six dataset, and independent of the number of components per partition ($64, 128, 256$), the NoFlo trick yields lower

---

[2]https://lightning.ai/

[3]The `cross` structure is taken from (Zuidberg Dos Martires, 2024), which they implemented for their experiments, even though they described the `neighbor` structure in the paper itself.

Table 2: Test set bpd for MNIST datasets (lower is better) using the `neighbor` structure and a 0.05 learning rate. The three columns show the bpd for different numbers of components per partition (64, 126, 256).We report the average over 5 runs and omit the variance as it was negligible. The bottom row shows the average difference (per data set) between using the NoFlo trick and LogSumExp trick.

| | 64 | | 128 | | 256 | |
|---|---|---|---|---|---|---|
| | NoFlo | LogSumExp | NoFlo | LogSumExp | NoFlo | LogSumExp |
| mnist | 1.49 | 1.52 | 1.46 | 1.51 | 1.44 | 1.50 |
| fmnist | 3.96 | 3.99 | 3.95 | 3.99 | 3.97 | 4.02 |
| emnist:mnist | 2.50 | 2.52 | 2.47 | 2.50 | 2.44 | 2.49 |
| emnist:letters | 2.54 | 2.57 | 2.51 | 2.55 | 2.48 | 2.53 |
| emnist:balanced | 2.59 | 2.62 | 2.56 | 2.60 | 2.53 | 2.58 |
| emnist:byclass | 2.49 | 2.51 | 2.44 | 2.47 | 2.40 | 2.43 |
| avg. improvement | 0.03 | | 0.04 | | 0.05 | |

Table 3: Test set bpd for MNIST datasets (lower is better) using the `cross` structure and a 0.05 learning rate. The three columns show the bpd for different numbers of components per partition (64, 126, 256). We report the average over 5 runs and omit the variance as it was negligible. The bottom row shows the average difference (per data set) between using the NoFlo trick and LogSumExp trick.

| | 64 | | 128 | | 256 | |
|---|---|---|---|---|---|---|
| | NoFlo | LogSumExp | NoFlo | LogSumExp | NoFlo | LogSumExp |
| mnist | 1.18 | 1.22 | 1.16 | 1.20 | 1.14 | 1.19 |
| fmnist | 3.39 | 3.43 | 3.35 | 3.40 | 3.34 | 3.39 |
| emnist:mnist | 1.72 | 1.76 | 1.66 | 1.71 | 1.62 | 1.67 |
| emnist:letters | 1.72 | 1.77 | 1.65 | 1.70 | 1.61 | 1.67 |
| emnist:balanced | 1.75 | 1.80 | 1.68 | 1.73 | 1.65 | 1.70 |
| emnist:byclass | 1.66 | 1.71 | 1.56 | 1.61 | 1.47 | 1.53 |
| avg. improvement | 0.05 | | 0.05 | | 0.05 | |

bpd than the LogSumExptrick. We repeated the experiment for varying learning rates using the `cross` structure and found similar behavior. These results are reported in Appendix C.

## 4.2 MEASURING THE COMPUTATIONAL COSTS

In Section 3.3 we gave a rough estimate of the computational cost based on the operations involved. Here we now measure them. To this end we measured the wall clock time needed for one training epoch on the MNIST dataset using the NoFlo trick and the LogSumExp trick, respectively. We report the measured times in Figure 3 (left). We also report in Figure 3 (right) the peak memory per epoch. As there were no significant differences between the methods for the latter we only plot the peak memory consumption for the NoFlo trick.[4]

In Figure 3 (left) we see that both methods follow a similar and almost linear scaling. This is due to the fact that the matrix-vector multiplications involved can be parallelized on the GPU to be performed in linear time. We also observe a constant off-set between the NoFlo circuit evaluations and the LogSumExpcircuit evaluations. This indicates that there is a fixed computational overhead for the NoFlo trick that can probably be improved upon with a more efficient implementation. As already mentioned, we did not see any significant difference between the memory footprints of both methods, cf. Figure 4 (right).

---

[4]To measure the memory we used the `max_memory_allocated` function from PyTorch https://pytorch.org/docs/stable/generated/torch.cuda.max_memory_allocated.html

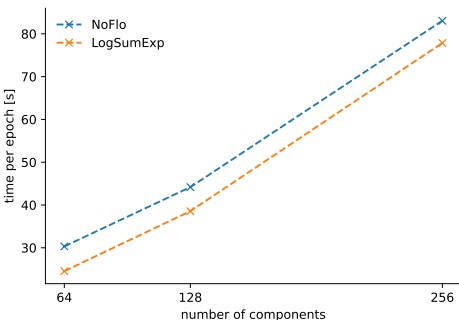 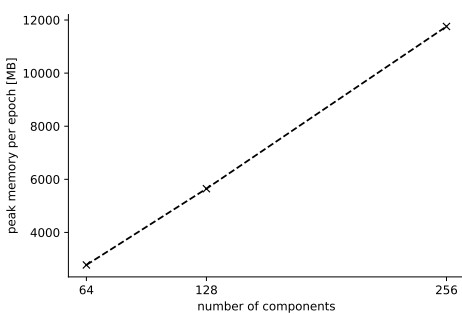

Figure 3: Left: run time per epoch in seconds for different numbers of components per partition. Right: peak memory usage for or different numbers of components per partition for circuit evaluations with the NoFlo trick. The `cross` circuit structure was used.

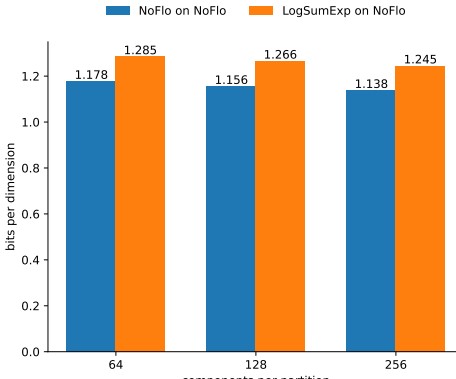 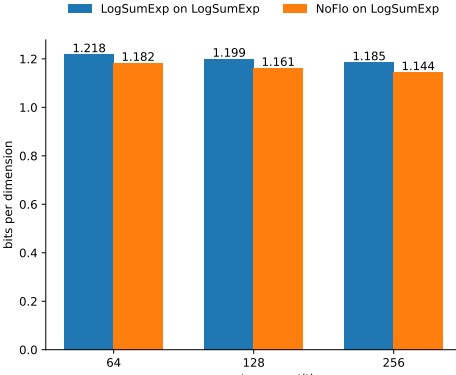

Figure 4: Bits per dimension on the test dataset for two models. Left: the model was trained using the NoFlo trick. Right: the model was trained using the LogSumExp trick. The legend "LogSumExp on NoFlo" means that we evaluate a model that was trained using the NoFlo trick by using the LogSumExp trick on the test set. Analogously for the remaining three legends.

### 4.3 A CURIOUS ABLATION STUDY

In order to find the root cause for the improved performance in terms of density estimation of the NoFlo trick over the LogSumExptrick we performed the following ablation study. We first trained a circuit (using the `cross` structure) with the NoFlo trick. On the test set we then evaluated the circuit with the NoFlo trick as well as with the LogSumExp trick. We performed this experiments on the MNIST dataset for varying number of components per partition. We report the results in Figure 4 (left). We also performed the analogous experiment where we performed the training with the LogSumExp trick and evaluated on the test set with both tricks. These results can be found in Figure 4 (right).

Inspecting the histograms in Figure 4 we observe some curious behavior. We do not only obtain lower bpd with the NoFlo vs. the LogSumExp trick, when comparing the blue bars from the left and right figures. We also obtain higher bpd when evaluating NoFlo-trained circuits using the Log-SumExp trick and lower bpd when evaluating LogSumExp-trained circuits using the LogSumExp trick.

Initially we suspected numerical precision issues to be the reason for this discrepancy between evaluations of one and the same circuits but with different numerical stable computation schemes. However, when repeating the experiment with $64$ bit precision instead of $32$ bits, this phenomenon still manifested itself.

As of now we are not certain where exactly the circuit evaluations with the LogSumExp trick lose the probability mass that leads to these lower bpd. We stipulate that it might be related to the repeated application of logarithms and exponentiation and that these operations are not exact reciprocals of each other when using finite precision. A counterargument to this thesis, however, is that increasing the precision from 32 to 64 should then have alleviated the issue. However, we were not able to observe such an improvement when increasing the precision.

## 5 CONCLUSIONS

In this paper we introduced normalized float trick, a new scheme to ensure numerically stable computations when evaluating probabilistic circuits. Experimentally we have shown that simply using the NoFlo trick instead of the standard LogSumExp trick leads to improved performance on suite of density benchmarks at very limited computational overhead. For practitioners this means that using the NoFlo trick constitutes a sensible implementation choice for boosting the performance of probabilistic circuits. However, it is also unclear where exactly this increase in performance exactly originates from. We leave the resolution of this open question to future work.

### REPRODUCIBILITY

We describe the NoFlo trick in pseudo-code in Algorithm 2 and have implemented it for probabilistic circuits in Python. We will release the source code upon acceptance. Our lab's policy is not to publicly share anonymous code repositories prior to acceptance. The code base will be made available to the reviewers through an anonymous link.

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

# A    PROOF OF PROPOSITION 2.3

**Proposition 2.3.** *Layered PCs are valid probability distributions.*

*Proof.* If $\pi_{root}(\mathbf{x})$ is the computation unit at the root of the layered PC, the unit forms a probability distribution if $\pi(x) \geq 0$, for every $\mathbf{x} \in \Omega(\mathbf{X})$ and if $\sum_{\mathbf{x} \in \Omega(\mathbf{X})} \pi(\mathbf{x}) = 1$. The first condition is trivially satisfied as the circuit only performs linear operations on matrices and vectors (one_hot($x_k$), $W_k$) with positive entries only. For the condition $\sum_{\mathbf{x} \in \Omega(\mathbf{X})} \pi(\mathbf{x}) = 1$ we observe that we can simply push down the summation for each variable to the respective leaf unit. This yields the following summation in the leaves:

$$\sum_{x_k \in \Omega(X_k)} W_k \times \text{one\_hot}(x_k) = W_k \times \sum_{x_k \in \Omega(X_k)} \text{one\_hot}(x_k) = W_k \times \mathbb{1}^M = \mathbb{1}^C. \qquad (25)$$

Here $\mathbb{1}^C$ denotes the $C$ dimensional vector having as entries only ones. For the last step in the equation above we exploited the fact that the weight matrices $W_k$ are row-normalized.

Passing on the marginalized leaves to the next layer gives us:

$$W_k \times \left( \mathbb{1}^C \odot \mathbb{1}^C \right) = W_k \times \mathbb{1}^C = \mathbb{1}^C \qquad (26)$$

Repeating this process until we reach the root will eventually result in $\sum_{\mathbf{x} \in \Omega(\mathbf{X})} \pi(\mathbf{x}) = 1$ and thereby also showing that the second condition is satisfied. $\qquad \square$

# B    LOG-PROBABILITIES AND THE LOGSUMEXP TRICK

The naive way of performing probabilistic inference is simple addition and multiplication of real-valued numbers from the unit interval $[0, 1]$. Unfortunately, due to the finite precision of physical machines, this leads to numerical stability issues in practice. The go-to technique to avoid such numerical stability issues is to map probabilities to log-probabilities:

$$\lambda = \log \pi \qquad (27)$$

where $\pi \in (0, 1]$ and $\lambda \in (-\infty, 0]$. We can now also map operations in linear space to adequate operations in log-space. For linear space multiplication we have:

$$\pi_1 \times \pi_2 = \pi_3 \Leftrightarrow \log \pi_1 + \log \pi_2 = \log \pi_3 \qquad (28)$$
$$\Leftrightarrow \lambda_1 + \lambda_2 = \lambda_3. \qquad (29)$$

In other words multiplication in linear space maps to addition in log-space. Moreover, for the addition we have:

$$\pi_1 + \pi_2 = \pi_3 \Leftrightarrow \log (\pi_1 + \pi_2) = \log \pi_3 \qquad (30)$$
$$\Leftrightarrow \log (\exp \log \pi_1 + \exp \log \pi_2) = \lambda_3 \qquad (31)$$
$$\Leftrightarrow \underbrace{\log (\exp \lambda_1 + \exp \lambda_2)}_{LSE(\lambda_1, \lambda_2)} = \lambda_3 \qquad (32)$$

We see that the log-space operation that is equivalent to addition in linear space requires first exponentiation, then addition, followed by taking the logarithm. This is often abbreviated as $LSE(\cdot, \cdot)$.

Unfortunately, exponentiation here is numerically unstable. In order to compute $\log (\exp \pi_1 + \exp \pi_2)$ in a numerically stable fashion the usual trick is to make use of the LogSumExp trick:

$$\log (\exp \lambda_1 + \exp \lambda_2) = \log (\exp(\lambda_1 - a + a) + \exp(\lambda_2 - a + a)) \qquad (33)$$
$$= \log (\exp(\lambda_1 - a) + \exp(\lambda_2 - a)) + a. \qquad (34)$$

Here $a$ is a constant and usually chosen to be $\max(\lambda_1, \lambda_2)$. The effect of subtracting $a$ from the log-probabilities before exponentiation is that we avoid numerical underflow issues and the computation of $LSE(\cdot, \cdot)$ becomes numerically stable.

# C  DENSITY ESTIMATION USING DIFFERENT LEARNING RATES

Table 4: Test set bpd for MNIST datasets (lower is better) using the `cross` structure and a 0.005 learning rate. The three columns show the bpd for different numbers of components per partition (64, 126, 256). We report the average over 5 runs and omit the variance as it was negligible. The bottom row shows the average difference (per data set) between using the NoFlo trick and LogSumExp trick.

| | 64 | | 128 | | 256 | |
| --- | --- | --- | --- | --- | --- | --- |
| | NoFlo | LogSumExp | NoFlo | LogSumExp | NoFlo | LogSumExp |
| mnist | 1.20 | 1.24 | 1.19 | 1.24 | 1.18 | 1.23 |
| fmnist | 3.43 | 3.47 | 3.42 | 3.47 | 3.42 | 3.47 |
| emnist:mnist | 1.76 | 1.79 | 1.72 | 1.76 | 1.70 | 1.75 |
| emnist:letters | 1.74 | 1.78 | 1.69 | 1.74 | 1.67 | 1.72 |
| emnist:balanced | 1.77 | 1.81 | 1.73 | 1.77 | 1.71 | 1.76 |
| emnist:byclass | 1.65 | 1.69 | 1.55 | 1.60 | 1.48 | 1.53 |
| avg. improvement | 0.04 | | 0.04 | | 0.05 | |

Table 5: Test set bpd for MNIST datasets (lower is better) using the `cross` structure and a 0.01 learning rate. The three columns show the bpd for different numbers of components per partition (64, 126, 256). We report the average over 5 runs and omit the variance as it was negligible. The bottom row shows the average difference (per data set) between using the NoFlo trick and LogSumExp trick.

| | 64 | | 128 | | 256 | |
| --- | --- | --- | --- | --- | --- | --- |
| | NoFlo | LogSumExp | NoFlo | LogSumExp | NoFlo | LogSumExp |
| mnist | 1.20 | 1.23 | 1.18 | 1.22 | 1.17 | 1.22 |
| fmnist | 3.42 | 3.46 | 3.40 | 3.45 | 3.39 | 3.44 |
| emnist:mnist | 1.74 | 1.78 | 1.71 | 1.75 | 1.69 | 1.73 |
| emnist:letters | 1.72 | 1.77 | 1.68 | 1.72 | 1.65 | 1.70 |
| emnist:balanced | 1.75 | 1.80 | 1.71 | 1.75 | 1.69 | 1.74 |
| emnist:byclass | 1.64 | 1.69 | 1.54 | 1.59 | 1.47 | 1.51 |
| avg. improvement | 0.04 | | 0.05 | | 0.05 | |

Table 6: Test set bpd for MNIST datasets (lower is better) using the `cross` structure and a 0.1 learning rate. The three columns show the bpd for different numbers of components per partition (64, 126, 256). We report the average over 5 runs and omit the variance as it was negligible. The bottom row shows the average difference between using the NoFlo and LogSumExp tricks.

| | 64 | | 128 | | 256 | |
| --- | --- | --- | --- | --- | --- | --- |
| | NoFlo | LogSumExp | NoFlo | LogSumExp | NoFlo | LogSumExp |
| mnist | 1.20 | 1.24 | 1.17 | 1.22 | 1.15 | 1.20 |
| fmnist | 3.43 | 3.47 | 3.40 | 3.45 | 3.38 | 3.43 |
| emnist:mnist | 1.74 | 1.79 | 1.68 | 1.73 | 1.64 | 1.69 |
| emnist:letters | 1.75 | 1.80 | 1.68 | 1.73 | 1.63 | 1.69 |
| emnist:balanced | 1.78 | 1.83 | 1.71 | 1.76 | 1.67 | 1.73 |
| emnist:byclass | 1.71 | 1.76 | 1.61 | 1.67 | 1.53 | 1.59 |
| avg. improvement | 0.05 | | 0.05 | | 0.06 | |

