# OpenReview forum: "The Normalized Float Trick: Numerical Stability for Probabilistic Circuits without the LogSumExp Trick"
_ICLR.cc/2025/Conference — Submitted to ICLR 2025_

### Official Review · Reviewer_sPEJ · 2024-10-21

**Soundness:** 2
**Presentation:** 2
**Contribution:** 1
**Rating:** 3
**Confidence:** 5

**Summary:**

The paper introduces the "Normalized Float Trick" (NoFlo), an alternative to the widely used LogSumExp trick for ensuring numerical stability when evaluating Probabilistic Circuits (PCs), as they could suffer from numerical instability when performing repeated multiplications of small values.
While the LogSumExp trick is the standard solution to mitigate this issue, the authors argue that the NoFlo trick leads to better density estimation with slightly less computational overhead.
The paper provides some explanations of the NoFlo trick and presents experimental results on the MNIST family of datasets, showing that models using NoFlo outperform those using LogSumExp in terms of bits per dimension (bpd) across various configurations.

**Strengths:**

- The introduction of an alternative to the well-established LogSumExp trick could indeed provide a fresh perspective on numerical stability in PCs

**Weaknesses:**

- There is lack of motivation. The paper does not provide a convincing explanation or theoretical justification for why the NoFlo trick should lead to improved performance over the LogSumExp trick. The claim that it "simply works better" seems insufficient, leaving the reader wondering why this new method is superior beyond empirical results.

- the analysis on the computational cost actually seems to suggest that there's should be practically very limited computational advantage in using the trick, as the cost is dominated by C^2, which can be very high for large models.

- comparison with very relevant literature is missing [1]

- I do also have the feeling that something may be off implementation-wise.  To check that models are actually encoding valid distributions, authors could build a small model for, say, 16 binary variables, compute all possible 2^16 log-likelihood and check if their sum is actually (very close to) zero. Could the authors try this test and report to me if the log-sum-exp implementation and the NoFlo return the same value? I'd also suggest to check what happens when playing with small models, without directly jumping to MNIST.

- I believe that the results reported in Fig 4 show that there's a bug somewhere as the difference in test-bpd between the LSE and NoFlo is too high to just be numerical precision. Playing with a toy dataset (as suggested above) might be insightful.


MINOR:
- The term "partition" in this paper is used in a different way than the common literature, and can therefore be misleading. Partition is in fact something usually referred to region graphs . Authors even cite relevant literature in this regard (see line 126), yet do not conform to the same language. For instance, the sentence "four partitions in the leaf layer" should just become "folded input layer", as authors seem to be somehow aware of folding techniques [2].

- Structured-decomposability is never formally defined, and when mentioned actually seems to refer to the simple "decomposability" (see L86-87)

- there's a typo in L155: what p_ki is?



[1] Yao, Lingyun, et al. "On Hardware-efficient Inference in Probabilistic Circuits." The 40th Conference on Uncertainty in Artificial Intelligence.

[2] Loconte, Lorenzo, et al. "What is the Relationship between Tensor Factorizations and Circuits (and How Can We Exploit it)?." arXiv preprint arXiv:2409.07953 (2024).

**Questions:**

- Why does the NoFlo trick improve performance? The paper lacks a clear theoretical justification for why the NoFlo trick yields better results than LogSumExp. Can the authors provide a deeper explanation for the mechanism behind this improvement?

- How does the NoFlo trick perform on models beyond PCs? While the focus is on probabilistic circuits, the NoFlo trick effectively be applied to any computation requiring the LSE.

- what does the sentence "Note that in the definition above we used Hadamard products to fold computations from the previous
layer" (L123-124) mean?

---

### Official Review · Reviewer_GwER · 2024-10-30

**Soundness:** 2
**Presentation:** 2
**Contribution:** 2
**Rating:** 3
**Confidence:** 3

**Summary:**

Probabilistic circuits use nested arithmetic operations with small values, naturally inducing numerical instability.
Practitioners usually employ the "LogSumExp trick" to mitigate this issue.
The authors propose an alternative to this method, the "normalized float trick".
They provide estimations of the computational cost of both methods, concluding that the normalized float trick is more computationally efficient.
Finally, they perform experiments on density estimation benchmarks and find that the proposed method brings consistent improvements over the classical one.

**Strengths:**

1. The base idea is simple, making the paper easy to follow.
    Less directly, the simplicity of the method could also lead to better adoption in practice and lead to more efficient implementations.

1. Section 3.3 (comparing the estimated computational cost) is a good start to the crucial task of comparing the efficiency of the two methods.

1. The authors seem to strive for clarity, which is always commendable.

1. The authors are upfront about important limitations of their work, such as the neglect of IO costs in section 3.3 and the current lack of understanding around the phenomenon discussed in section 4.3.

**Weaknesses:**

In summary, while I appreciate the overall idea, I believe this work needs to improve in several aspects (listed below) to be considered for publication (especially at such a prestigious venue).
I highlight that I see potential in this research, in that it would be particularly strengthened by addressing some key limitations.

1. At line 453 the authors write "The code base will be made available to the reviewers through an anonymous link".
    Unless I somehow missed it, this is not true.
    Access to the source code can sometimes be very useful to the review process and denying it asks for a better justification than "lab's policy".

    **[Extra]** If such a policy is as unique as it seems to me, disclosing it may compromise the authors' anonymity to some extent (consider the extreme case where a single lab has such a policy and one of the reviewers knows that to be the case).
    The authors should be mindful of the potential issue.

1. No train-data performance is reported even though overfitting might be an important factor in the results.
    Without this information, one cannot disregard the possibility that the proposed method leads to **higher** numerical distortions compared to the LogSumExp trick, since this error could lead to some sort of regularization that would explain the improvements in test-data performance detected by the experiments.
    Put differently, it seems like a reasonable hypothesis that, by erasing information/adding noise, underflowing reduces the expressiveness of the model, which could lead to better generalization.
    This hypothesis seems worth of some discussion, at least.

    See Question 1.

1. Overall, the work lacks a deeper, more meaningful comparison with the reference method.
    When starting to read the paper, I was expecting to soon find some rich theoretical analysis of the numerical stability of the methods, perhaps with results bounding the error propagation in terms of the weight matrices (their spectral norm, I guess).
    Instead, the closest to that would be the estimations in section 3.3, which is a nice starting point but cannot be generalized too much given how complex the computational cost of low-level methods can become. (The authors seem aware of the difficulty, as they, e.g., hint at the significance of hardware particularities).

    Without a solid theory, generalizing the performance of the proposed method to other settings depends heavily on the experiments.
    However, those are too limited to fulfil this role.

    In summary, while I do not doubt it, I remain unconvinced of the generality of the results presented, and, thus, of the significance of the proposed method.

1. Given the striking analogy between the proposed method and the floating point representation used in hardware, it is surprising that the authors do not discuss any relationship between the two at all.
    For example, in terms of floating point representation, the proposed method could be loosely seen as increasing the precision in the representation of $\pi \in [0, 1]^d$ by storing an adaptively chosen common offset for the exponent (a.k.a. characteristic) of the representations of the coordinates of $\pi$, with this offset being stored in $\beta$.
    (Here, I ignored the significand (a.k.a. mantissa) for simplicity.)

    I am not sure to which extent issue is pertinent, so I will regard it as a minor one.
    Still, I recommend the authors consider it.
    For instance, I would not be surprised if investigating this direction led to some useful "bit-hacks" that improve the efficiency of the proposed method.

1. It is somewhat odd that section 3.3 eventually uses cycle-counts estimated for x86 architecture while experiments are run on GPUs.
    This inconsistency further reduces how much we can learn from the estimations in that section.

1. The writing is subpar.
    1. There are many typos.
    1. The overall text feels artificially stretched, with long phrasings, repetitions, and an overall slow pace that reduces clarity rather than increasing it.
    1. The mathematical writing is not tight enough. The authors frequently fail to fully characterize mathematical objects (usually by omitting domains of index variables) and typos are also present in the maths.
    1. Motivation, intuition, or references are missing around some important design choices.
        Without those, passages like paragraphs 286-290, 297-302, and 308-311 are hard to appreciate.
        Why those choices? What were the alternatives? How do they compare? What should be their positive and (more importantly) negative implications?

        An example would be the excerpt "For the sake of exposition we will limit ourselves to canonical polyadic layers and refer the reader to (Mari et al., 2023) for alternatives" (around line 126).
        It is fair and a good start for the authors to make this comment.
        However, this reference still asks for some discussion (even if brief) on how significant those alternatives are and how the authors' contribution relates to them.

        Even arbitrary choices need to be properly signalled as such to allow for proper interpretation of the results.


---
### Minor issues and suggestions

1. The times reported in Figure 3 seem unusually long.
    Are you sure everything is consistently running on the GPU?

**Questions:**

In your experiments, what do the test and train performances look like across epochs?
Do the improvements detected also apply to the train-data performance?

---

### Official Review · Reviewer_zAWd · 2024-11-02

**Soundness:** 2
**Presentation:** 2
**Contribution:** 3
**Rating:** 5
**Confidence:** 4

**Summary:**

The paper introduces a new approach to evaluate and perform probabilistic inference with probabilistic circuits in a numerically stable way. The proposed approach -- called NoFlo trick -- can substitute the ubiquitous log-sum-exp trick while delivering significant computational speed-ups with a negligible memory increase. Experimentally, the NoFlo trick is found to apparently improve the performances of probabilistic circuits for distribution estimation tasks.

**Strengths:**

I believe the proposed trick to evaluate probabilistic circuits (NoFlo trick) is a significant contribution for the following reasons:
(1) it has the potential to replace the log-sum-exp trick, which has been the standard for several years to evaluate not only circuits but also graphical models.
(2) it is surprisingly simple and can be easily vectorized to be run on the GPU, in the same way as the log-sum-exp trick.
(3) it brings a significant computational speed-up with (almost) no memory increase, by avoiding the evaluation of exponential functions.
For these reasons, I think this paper could have a considerable impact on the implementation of many papers about probabilistic circuits.

I have appreciated the inclusion of an experimental section comparing the performances for distribution estimation tasks (NoFlo vs log-sum-exp trick). The reason is that it uncovered the possibility of improving the performances of probabilistic circuits by simply changing how layers are evaluated. To the best of my knowledge, this is a novel discovery and deserves future investigation, as it may impact the experimental results of other papers as well.

About the presentation, overall I found the paper quite easy to read, with a simple and concise language.

**Weaknesses:**

Regarding the presentation, I think the paper has some weaknesses as it introduces a different terminology from some concepts that are already known. I believe these can be addressed during the rebuttal.
(1) The paper mention the concept of partition tree from a very recent paper (L097). However, the same definition looks the same of vtree [A] or pseudo tree [B].
(2) The paper defines partition circuits (Defn. 2.1.) and layered probabilistic circuits (Defn. 2.2.). The authors might be interested in unifying the terminology with the definition of tensorized circuit appearing in a follow-up work by Mari et al. 2023 [C].

[A] Pipatsrisawat, K.; and Darwiche, A. 2008. New Compilation Languages Based on Structured Decomposability.
[B] Dechter, R.; and Mateescu, R. 2007. AND/OR search spaces for graphical models.
[C] L. Loconte, A. Mari, G. Gala, et al. 2024. What is the Relationship between Tensor Factorizations and Circuits (and How Can We Exploit it)?

In the abstract and introduction of the paper, the authors mention that the NoFlo trick boosts performances of probabilistic circuits for distribution estimation. However, I do not believe this is a claim supported by the evidence. Although the paper reports a decrease in bits-per-dimension (BPD) when using the NoFlo trick either at evaluation or training time, this decrease cannot be interpreted as a significant improvement. The reason is that the "evaluation ground truth" -- the true BPD scores one would get with infinite precision -- is not known, and it could be closer to the one achieved by using the NoFlo or log-sum-exp trick. As such, the performance boost that has been observed empirically could be due to some additional unknown numerical instability occurring in the NoFlo trick.

The paper lacks experiments on smaller scales. By scaling the size of the circuits, it becomes more difficult to understand which evaluation method is more numerically stable and therefore determine which BPD is closer to the "ground truth" (see above). I think the paper would be much more solid if it provided results comparing three evaluation methods: NoFlo trick, log-sum-exp trick, and evaluation with no tricks (i.e., by evaluating sum and product as is). By performing these experiments on much smaller circuits and in float64 (thus avoiding underflows), the authors might be able to determine whether the differences in BPD is due to other types of numerical instabilities occurring in very large circuits only.

The two points mentioned above bring my score substantially down. However, I am open to increase it if further evidence is provided.

**Questions:**

In section 4.3. the authors perform an ablation study about the performances gain achieved by using the NoFlo trick. In addition, the authors mention they tried using 64-bits floating-point arithmetic with no change in the results. Could the authors provide a figure similar to Figure 4 as to support this claim when using 64-bits precision?

I did not find an implementation (e.g., in PyTorch) attached to the paper submission. I find this to be a negative aspect of the submission. Could the authors please redirect me to it?

---

### Official Review · Reviewer_iN1g · 2024-11-10

**Soundness:** 2
**Presentation:** 2
**Contribution:** 2
**Rating:** 3
**Confidence:** 5

**Summary:**

This paper propose a normalized float trick to replace the logsumexp trick in probabilistic circuits evaluation and training. In order to mitigate the underflow issue, the normalized float trick carries normalization constants during the forward and backward pass of probabilistic circuits. In experiments section, the author demonstrates that this method can improve performance.

**Strengths:**

The proposed method of replacing logsumexp with normalized float trick is interesting, the main idea is to carry the $\alpha_k$ terms in equation 6 during training and evaluation.

**Weaknesses:**

1. The authors claim the proposed method is more numerically stable than logsumexp, however, they do not verify or explain why this trick is more numerically stable. My understanding is that the proposed method is to cache the $\alpha_k$ terms during training and evaluation, but it’s unclear how this leads to improved stability. Is the stability enhancement due to reduced log and exp computations on $\alpha_k$? Comparing likelihoods would make this claim convincing.

2. In the experiments, the authors implement both logsumexp and nofloat trick to compare their performance, why not compare the log-likelihoods with existing methods, such as [1]? Is there still a performance improvement there. The architecture and training pipeline are bothn implemented by the authors themselves, why not use existing library, such as Peharz et al. or 2020; Liu et al., 2024; to the best of my knowledge, the existing libraries handle numerical stability very well. Is there still performance improvement if you use existing library.

3. The writing is not clear enough. For instance, is Proposition 2.3 based on prior work, and what role does it play in the overall argument?

4. The authors do not seem to be familiar with the related work, for example, in line 78, the layered circuit is not introduced in Shih et al. (2021). What Shih et al. (2021) does it to use neural models for parameter sharing and thus regularization. The architecture is commonly used and introduced in [2] and Peharz et al. or 2020.

[1] Pedro Zuidberg Dos Martires. Probabilistic Neural Circuits. AAAI 2024.
[2] Robert Peharz, Antonio Vergari, Karl Stelzner, Alejandro Molina, Martin Trapp, Xiaoting Shao, Kristian Kersting, and Zoubin Ghahramani. Random sum-product networks: A simple and effective approach to probabilistic deep learning. UAI, 2019.

**Questions:**

1. The author introduce Partition Circuit first and then the NoFloat trick, is the trick only applied to Partition Circuit?

2. The log-likelihood improvement in Table 2 seems too good to be true. Can you explain where does this improvement come from? If you train a circuit with fully-factorized distribution and the architecture only has one layer, does it still have similar performance improvement.

---

### Meta-Review · Area_Chair_1QH9 · 2024-12-19

**Metareview:**

The paper 'The Normalized Float Trick: Numerical Stability for Probabilistic Circuits without the LogSumExp Trick' was reviewed by 4 reviewers who gave it an average score of 3.5 (final score: 3+3+3+5). The reviewers found multiple issues with this submission. Even if the idea is appreciated, the presentation and linking to current knowledge in the field are far from what is expected from an ICLR paper. All reviewers recommend rejecting the paper in its current form.

**Additional Comments On Reviewer Discussion:**

The authors did not provide a rebuttal, and thus no further interaction between the authors and reviewers was expected.

---

### Decision · Program_Chairs · 2025-01-22

Reject